# Effects of announcing a vocabulary test before reading a glossed text on reading behaviors and vocabulary acquisition: An eye-tracking study

Hayoung Kim[1], Sungmook Choi[1], Soo-Ok Kweon [2]*

1 Department of English Education, Teachers College, Kyungpook National University, Daegu, South Korea,
2 Division of Humanities and Social Sciences, Pohang University of Science and Technology, Pohang, South Korea

* soook@postech.ac.kr

**Data Availability Statement:** All relevant data are within the paper and its Supporting information files.

## Abstract

Glosses provide an effective way of fostering second language (L2) vocabulary acquisition. Expanding on previous research, we explored how a vocabulary test announcement prior to reading a glossed text influences the reading behaviors and subsequent vocabulary acquisition of L2 learners. The participants of this study comprised 65 Korean undergraduate students. The participants were assigned to either a vocabulary test announcement (TA) or a no test announcement (NTA) group. Thereafter, the researchers read to the participants a short story containing 16 glossed words displayed in the bottom margin. The students' responses to the reading comprehension test and a battery of vocabulary tests (i.e., form recall, meaning recall, and meaning recognition) were captured immediately and one week after the assessments. The results showed that while processing *in-text* target words and marginal glosses, no variation in eye-tracking measures (e.g., gaze duration and total reading time) was observed among the TA and NTA groups. In contrast, the TA group spent significantly longer time fixating on *bottom-margin* target words than the NTA group during eye-tracking measures. Regardless of the testing phase, the vocabulary test results showed that only the form recall scores in the TA group were significantly higher than the NTA group. However, the differences in meaning recall and recognition scores were not significant. Collectively, these results suggest that vocabulary test announcements likely enhance the favorable effects of glossed text, particularly through promoting visual word form acquisition. However, the effects do not strengthen form-meaning associations without compromising L2 learners' reading comprehension.

## 1. Introduction

A gloss is a definition or short explanation of an unfamiliar word in a text. Recently, several researchers have shown that glosses can facilitate *incidental* vocabulary learning among second

**Funding:** The authors received no specific funding for this work.

**Competing interests:** The authors have declared that no competing interests exist.

language (L2) learners. Previous studies [1, 2] have suggested that reading glossed text promotes vocabulary learning in contrast to reading non-glossed text. For example, a meta-analysis by Yanagisawa et al. (2020) [2] found that learners who read glossed texts learned more of the unknown target words (45.3% and 33.4%, respectively, for immediate and delayed post-tests) than learners who read non-glossed texts (26.6% and 19.8%, respectively).

Studies have also investigated the effects of glosses on *intentional* vocabulary learning, which is distinct from *incidental* vocabulary learning, by assessing whether announcing a vocabulary test in advance can affect whether new words are retained [3–7]. "Test announcement" refers to an early warning informing learner to perform the required vocabulary tests after reading. Consequently, this announcement can influence learners to increase the amount of attention devoted to the glossed target words while reading a glossed text, thus strengthening the connection between the visual word form and meaning (e.g., apple-사과) of target words. However, studies are yet to examine whether test announcements actually induce learners to increase the amount of attention they pay to the target words; that is, whether learners devote additional cognitive effort to the target words. Moreover, few studies have assessed the relationship between cognitive effort and vocabulary acquisition.

Thus, to address this research gap, the present study used both offline (paper-based tests) and online (eye-tracking) methods to answer the research questions proposed in this study. The results of this study are likely to clarify the effects of glosses on incidental and intentional vocabulary learning. Particularly, the current study aims to examine the trade-off effect between vocabulary learning and reading comprehension. This is based on the possibility that prereading test announcements have the potential to increase the effort devoted to processing the target words while decreasing reading comprehension as a result.

The research questions of the present study are formulated as follows:

1. What are the effects of a test announcement on learners' attention?

2. What are the effects of a test announcement on learners' vocabulary acquisition (i.e., form acquisition and form-meaning association)?

3. What are the effects of a test announcement on reading comprehension?

The first research question focused on how participants use "attention" as a result of test announcements. In the present study, attention can be defined as the time spent by learners reading the three regions of interest (ROIs): in-text target words, bottom-margin target words, and glosses. The in-text target words are presented in the text and the bottom-margin target words occur at the bottom of the text. Both correspond to the 'form' of a word which contains the information of spelling or pronunciation [8]. When learners look at the in-text or bottom-margin target words, they may acquire the orthographic lexical form of the words (i.e., fusk, tarb). On the other hand, glosses enable learners to obtain semantic information, that is the 'meaning' of words (i.e., 소음, 가방). It may encourage learners to activate the form-meaning links (i.e., fusk-소음, tarb-가방). Thus, the reading time that the participants spend gazing at the 'in-text target words' and 'bottom-margin target words' can be related to the acquisition of 'form' of words, and the time spent reading the 'glosses' can be related to the 'form-meaning connection' of words. To quantify the focus placed on these three ROIs among readers, we used three measures of eye movement: gaze duration, second-pass reading time, and total reading time. "Gaze duration" (also known as first-pass reading time) represents the sum of all fixations on an ROI before moving to another ROI, and describes the initial processing of word recognition (i.e., early measure). Furthermore, "second pass reading time" indicates the sum of all fixations in an ROI following the initial first-pass fixations. "Total reading time" defines the sum of all fixations made in an ROI. Research suggests that both the total reading

time and second-pass reading time reflect higher-level processing (i.e., late measure), such as the discourse-level processes of text integration. In addition to these three measures, the fixation count was also measured, which refers to the number of fixations made within an ROI. Among these eye-tracking measures, total reading time is considered critical as it can be used to quantify the attention that a reader pays to an experimental passage [9].

The second research question addressed the involvement of "vocabulary acquisition," which is measured by form acquisition and the form-meaning connection of target words. This acquisition is reflected by the scores on vocabulary tests in which participants assess the knowledge of target words after reading an English passage. Form acquisition estimates participants' ability to recall the form (spelling) of the target words in the form recall test. A form-meaning connection occurs when participants recall or recognize the meaning of the words provided and is measured using the meaning recall and meaning recognition tests. These vocabulary tests were administered both immediately (short-term test) and one week (long-term test) after reading the text. In the present study, "reading comprehension" is determined by the scores obtained on the post-reading reading comprehension test.

While glosses enable access to the meaning of new words, limited few studies have explored how glosses help students connect the form and meaning of new words when participants' attention to the vocabulary increases in response to a test announcement prior to reading a text. Thus, this study intends to extend research underscoring the effects of glosses and the effect of vocabulary test announcements on their benefits. For these purposes, both online (eye-tracking) and offline (vocabulary tests) measures were used.

## 2. Literature review

### 2.1. Incidental and intentional vocabulary learning

*Incidental* vocabulary learning describes the acquisition of vocabulary items "incidentally" as an outcome of the reading, listening, speaking, or writing activity [10]. Particularly, incidental learning of vocabulary requires repetitive exposure to new words through extensive reading or listening. Furthermore, this learning likely occurs when a learner infers the meaning of unknown words correctly from the context. Several researchers have insisted that incidental vocabulary learning is more effective for long-term memory than intentional vocabulary learning [11, 12]. However, since incidental vocabulary learning requires frequent exposure to language input, it is not easy for learners to be exposed to the same words repeatedly while reading books or watching TV, and it may take a long time to succeed in learning words. Moreover, the wrong meaning of words could be learned, if learners are not provided with correct cues that help them retrieve the meanings of unknown words from context [13, 14].

In contrast, *intentional* vocabulary learning refers to all language activities that are conducted to attain vocabulary. For example, Webb (2007) [15] defined intentional vocabulary learning as completing activities explicitly designed for word learning, such as crossword puzzles, word cards, or word lists with synonyms and antonyms. These decontextualized activities are prone to lead learners to rote learning. However, Hulstijn (1992) [13] believed that intentional vocabulary learning can also occur when learners engage in contextualized activities such as reading or listening when they are aware of an upcoming vocabulary post-test. In this vocabulary learning situation, the presence or absence of the test announcement is considered an important parameter. Thus, the operational definition of intentional vocabulary learning in this study is informed by Hulstijn (1992) [13], which is determined by the presence or absence of a test announcement.

## 2.2. Glossing and incidental vocabulary learning

Many researchers have regarded glossing as a useful tool to address the shortcomings of incidental vocabulary learning from a simple reading. Furthermore, glossing is considered a type of input modification that can increase the effect of vocabulary learning. Particularly, a gloss refers to a short definition or explanation of an unfamiliar word in a text and is understood to help learners' reading comprehension and vocabulary acquisition [14, 16]. Learners can gain a new vocabulary word by simply reading L2 text but providing gloss can further promote the vocabulary learning process. This is because gloss may allow learners opportunities to notice the lexical items in the reading materials and also to figure out the exact meaning of the new words, eliminating the risk of inferring the wrong meaning.

Numerous studies have examined the effects of glossing on incidental vocabulary learning. For instance, Hulstijn et al. (1996) [17] examined 78 advanced L1 Dutch learners of L2 French. The participants were required to read a short story under one of three conditions: marginal gloss (MG), dictionary (D), and control (C). This research compared incidental vocabulary learning in the MG group with that in the D and C groups. The text that was read comprised a short story written in French. Furthermore, the text comprised 1,306 words containing 16 unfamiliar target words, with 8 words appearing once (F1 words) and three times (F3 words), respectively. For participants in the MG group, the 16 target words were printed using a bold font and an L1 Dutch translation was provided on the right-hand margin. The marginal glosses for F3 words were provided once on their first occurrence. After reading the text, the participants were administered three kinds of vocabulary tests for the 16 target words: a form recognition test, a meaning recall test, and a meaning recall test for the context-specific words. The results exhibited better acquisition of form and meaning of the target words among the MG group in contrast to the D and C groups. Furthermore, Ko (2012) [18] compared the effects of glosses in incidental vocabulary learning with 90 Korean college students studying English as a Foreign Language (EFL) who were divided into groups and assigned to one of three conditions: no gloss, L1 gloss, and L2 gloss. The results indicated that the gloss group recognized approximately five or six more words than the no gloss group on a meaning recognition multiple-choice test. Moreover, participants gained word meaning for an average of 13.48 words, 12.86 words, and 7.55 words in the L1 gloss condition, L2 gloss condition, and no gloss condition, respectively. Thus, these results suggest that glossing can improve vocabulary acquisition.

Furthermore, many studies have demonstrated the effects of various kinds of glosses on vocabulary learning. For example, Yanagisawa et al. (2020) [2] conducted a meta-regression analysis of 42 research papers on glossing with respect to the *language* of glossing (L1/L2), *type* of glossing (interactive/non-interactive), and *mode* of glossing (pictorial/textual/audio/video). The results indicated that L1 glossing was more effective than L2 glossing. Furthermore, the study found that interactive glossing (e.g., hyperlinked glosses, multiple-choice glosses) was more effective than non-interactive glossing for in-text glosses. Moreover, interactive glosses, such as hyperlinked glosses, require learners to take action by clicking on a word. Therefore, the participants had time to retrieve the meanings of words before seeing the glosses. This retrieval of word meaning potentially involves cognitively demanding processing, which makes interactive glosses more effective than non-interactive glosses in vocabulary learning. Finally, no significant differences were observed among the glossing modes in the study.

Recently, eye-tracking studies have also been conducted to investigate glossing effects on vocabulary learning. Kang et al. (2020) [19] used eye-tracking to examine the contributions of L1 and L2 glosses to attention and vocabulary acquisition. The participants comprised 81 Korean university students who were randomly assigned to groups that were asked to read the

same passage with either L1 gloss (Korean), L2 (English), or no gloss. The eye-tracking analyses showed that both L1 and L2 gloss groups spent more time than the "no gloss" group on reading target words at the bottom margin, which implies glossing may facilitate learners' attention to the words. Moreover, the vocabulary test results indicated that L1 and L2 glosses were not effective in learning the word form but had a significant effect on reinforcing the form-meaning association of vocabulary.

Previous research has shown that glosses are useful for learning vocabulary while reading texts [17]. Thus, various attempts have been made to increase the effect of glosses by differentiating their language or type [1, 2, 18]. In addition, a recent study using eye-tracking [19] provided quantitative evidence that glosses increased readers' attention to new words while reading.

## 2.3. Glossing and intentional vocabulary learning

Many studies have quantified the glossing effect in *incidental* vocabulary learning. However, few studies have underscored the effect of glosses in the context of *intentional* vocabulary learning [3–7]. In the context of the current study, intentional vocabulary learning refers to a vocabulary test announcement received by learners before reading a text stipulating that they will be tested after reading the text [10]. This announcement could influence learners to enhance their focus on unfamiliar or glossed words. In contrast, the absence of an announcement creates a condition where learners could overlook the words. Thus, the test announcement can promote *intentional* vocabulary learning.

Although intentional vocabulary learning implies more effective word acquisition by allowing learners to pay more attention to the glossed words, there have been few studies on intentional vocabulary learning [3–7]. Moreover, they have used different experimental designs, making it difficult to compare findings.

For instance, Ko (1995) [3] used paper-based test to examine the effects of test announcements on vocabulary learning. A total of 189 Korean university EFL students were randomly assigned to three test announcement (TA) groups (i.e., L1 gloss, L2 gloss, no gloss) and three no test announcement (NTA) groups (i.e., L1 gloss, L2 gloss, no gloss). Subjects were asked to read short stories (854 words) printed on paper according to the gloss conditions of their group (L1, L2, no gloss) and complete a multiple-choice vocabulary test (meaning recognition test) immediately and one week later. The results indicated higher average scores among the TA group, i.e., intentional vocabulary learning group, than the NTA group, i.e., incidental vocabulary learning group for both immediate and delayed vocabulary tests, regardless of the language of the gloss (L1/L2). However, the difference was not significant.

On the other hand, some studies used computer-based assessment to explore the effectiveness of test announcement. Peters (2007) [5] presents the text on a computer screen that required participants to click on the unknown words to check their glosses. When they click on a word, a gloss appears in a pop-up window on the screen. In this study, 84 Dutch university students who randomly assigned to either the TA or NTA groups were asked to read a German text (1,096 words) containing 16 target words. While reading, the number of times they check the glosses was recorded automatically and this look-up behavior was used to quantify their attention to the vocabulary items. After reading the text, they were asked to respond to four kinds of vocabulary test: form recognition, meaning recall, contextual meaning recall, and meaning recognition. As a result, the TA group clicked more times to check the gloss but did not show higher vocabulary acquisition in four kinds of vocabulary test than the NTA group. In Peters' (2007) [5] study, test announcements had no positive effects on vocabulary learning. In contrast, Peters et al. (2009) [6] showed positive effects of test announcement in a

similar experimental design with Peters (2007) [5]. 137 Dutch-speaking university students read a German text (1,096 words) on a computer screen and they can refer to the gloss of the 16 target words at the click of the word. This study revealed that test announcement not only increased the number of times learners checked glosses but also improved their form recognition of the target words. Likewise, Zandieh and Jafarigohar (2012) [7] found the TA group showed better immediate retention of the word meaning than NTA group. 184 learners of English were divided into TA and NTA groups and read text with hypertext glosses on a computer screen. Hypertext glosses provide a definition of hyperlinked glossed words at the bottom of the screen. The TA group, under the intentional learning conditions, performed better on recognizing the meaning of words immediately but showed dramatic decrease of retention rate on delayed test.

A recent study recorded eye movements to analyze how test announcement affect learners' attention and vocabulary uptake from audio-visual input. Montero Perez et al. (2018) [4] investigate the effect of test announcement and types of subtitle using a 2 (TA or NTA) × 4 (type of captioning: no captioning, full captioning, keyword captioning, and glossed keyword captioning) between-subject design. A sample of 227 Dutch-speaking university students was divided into eight groups and asked to watch a French (L2) video under one of the eight stipulated conditions. Among the eight groups, the glossed-keyword-captioning groups were able to check the gloss by pressing the space bar while watching the captioned video. The result showed that the glossed-keyword-captioning groups achieved the highest score on the form recognition and meaning recall vocabulary test. This indicates that providing access to meaning of words through glossing can promote higher vocabulary uptake from video. As for the test announcement, it did not have any significant effect on learners' vocabulary acquisition or attention to the words. The authors assumed that these findings may have been influenced by the experimental design. After viewing each video clip, participants were asked to complete the comprehension tasks, but the vocabulary tests were administered only after watching all three video clips. Thus, it may have led participants to focus more on understanding content rather than on learning new vocabulary. The questionnaire results also suggest both TA and NTA groups focused primarily on the meaning of unknown words to answer the comprehension tasks rather than on the form of words.

The results of these studies on the effects of test announcements in glossed text lack consistency. Some studies have shown that test announcements have no effect on vocabulary acquisition [3–5], whereas others have found positive effects [6, 7]. In addition, the results differed depending on the kind of vocabulary test. For example, Peters et al. (2009) [6] found that the test announcement only facilitated recognizing the form of the target words. However, Zandieh and Jafarigohar (2012) [7] found that the test announcement helped recall the meaning of vocabulary. Furthermore, it was difficult to compare findings because previous studies used different experimental designs. Ko (1995) [3] used experimental text printed on paper, whereas Peters (2007) [5], Peters et al. (2009) [6], and Zandieh & Jafarigohar (2012) [7] presented online text through a computer screen that required participants to click on glosses represented by a pop-up dictionary. Moreover, Montero Perez et al. (2018) [4] incorporated glossed-captioned video viewing.

## 3. Method

### 3.1. Participants

This study comprised 65 undergraduate students (22 men and 43 women) from a Korean university. The participants were all native Korean speakers aged 20–27 years ($M$ = 23.29) and were considered intermediate to high-intermediate learners of English (L2) based on their

scores on the Nelson-Denny Reading Test. Similar to the 9th-grade students in the United States, the mean score of the participants was 53.32 out of 76 points (highest = 72; lowest = 30). All participants voluntarily applied through an online application form posted on the university's student website. Upon completing the online application form, the students provided their informed consent to participate in the research by providing their demographic information. Particularly, the consent form stated that students ought to only provide their demographic information if they agreed to participate in the study. Prior to conducting the experiment, an institutional review board (IRB) entitled the Bioethics Review Committee of Kyungpook National University approved this study (approval no.: KNU-2021-0176).

The exclusion criteria comprised individuals who had very poor eyesight or a physical or mental illness, as the experiment required participants to read the text through an eye-tracker. Participants were briefed about the study before participating in the experiment. However, the volunteers were only informed of the original purpose of the study after concluding the experiment.

### 3.2. Design

This study applied a between-group design, using the presence or absence of a vocabulary test announcement as the independent variable. This design allowed us to distinguish between intentional and incidental vocabulary learning groups [10]. The TA group, which represents an intentional learning condition, was notified in advance about the reading comprehension test and vocabulary test before reading the text. However, the NTA group, which represents an incidental vocabulary learning condition, was told only about the reading comprehension test, and thus was not prepared for the vocabulary test. Participants were randomly assigned to the TA ($n$ = 33) and NTA groups ($n$ = 32). Consequently, the groups were homogeneous in age ($t_{63}$ = 0.780, $p$ = 0.438, $d$ = 0.193) and English reading proficiency ($t_{63}$ = 0.773, $p$ = 0.442, $d$ = 0.192), as measured using the Nelson-Denny Reading Test.

### 3.3. Materials

**3.3.1. Experimental text.**   A short story titled "The Little Hunters at the Lake" from a graded reader at Oxford University Press was used as the experimental text in this study. The story originated in Turkey, and it tells the story of young boys hunting a bird that symbolizes "eternal love," which even veteran hunters never kill. The story provides a lesson through the noble love of birds and has an eventful plot, thus facilitating the researchers in evaluating the participants' understanding of the text. Furthermore, the text could be understood without prior knowledge. Reading comprehension can be affected by readers' prior knowledge. Therefore, it needs to be controlled for in the experiment [20].

In terms of linguistic elements, the text consisted of 1,628 words, of which 16 target words appeared 48 times, accounting for approximately 3% of the total text. The readers would experience challenges with understanding the content if more than 5% of unfamiliar words appeared in the text [21]. Thus, we tried to select a text composed of words that the participants would be familiar with. According to the Flesch-Kincaid Grade Level, the readability of the text was Level 2.1, indicating that a native speaker with two years of formal education can read the text without much difficulty. In the context of Korean college students, a text with this level of vocabulary was considered appropriate as an experimental text. Finally, we modified the names of characters from the original story. Particularly, the Turkish names of the characters were changed to English names to enhance readability for the participants. For example, "Hikmet" was replaced by "Hudson," and "Halil" was replaced by "Morgan." Since this study is about vocabulary learning that occurs during text reading, we wanted to avoid presenting

new or unknown words as much as possible other than target words in the text. Turkish names such as "Hikmet", "Halil", and "Tekin" would be less familiar to EFL Korean learners than English names and may be considered new words by them. Therefore, we modified the Turkish names to the English names to prevent participants from recognizing those words as target words, so that the purpose of this study could be more clearly examined.

**3.3.2. Target words and glosses.** The experimental text contained 16 target words, which were classified into two sets of target words: (1) eight words that appeared twice (F2 words) and (2) eight other words that appeared four times (F4 words). We chose F2 or F4 words as target words from the reading text, following Choi (2016) [1]. It was extremely difficult to find target F2 or F4 words that belong to the same part of speech, because glosses are used for words of different part of speech in the text to which L2 learners are exposed. Both sets of target words consisted of four nouns, four verbs, four adjectives, all of which were content words. The fact that all target words were content words might have facilitated participants' accurate and fast comprehension of the text. However, it seems to be impossible to objectively measure to what extent they are important. These 16 target words were replaced by pseudowords extracted from the ARC Nonword Database. These pseudowords did not violate the phonotactic rules of English though they do not exist [22]. This study used pseudowords as target words because the subjects' prior vocabulary knowledge might have affected the experimental results [1, 15].

The target words that appeared in the text were presented with glosses at the bottom margin of the text (Fig 1). Glosses comprised short definitions or synonyms of the target words in Korean, which was the participants' first language (L1). Several studies have demonstrated that L1 glosses are more effective than L2 glosses in helping students acquire the meaning of

The sky above the little lake was hule of birds — small birds, big birds, birds of all colours. We sat in the rain by Hudson's garden wall and watched them.

"Dizz Winter's coming," I said to my friends.

"The birds are beginning to leave and fly away to warm countries."

Then a hunting dog came by. It stopped and smelled all of us, then went away.

* hule: 가득찬     * dizz: 추운

**Fig 1. Eye-tracking stimuli sample.**

**Table 1. Summary of target pseudowords and glosses.**

| Target pseudowords | Glosses | Original words |
|---|---|---|
| fusk | 소음 | noise |
| tarb | 가방 | bag |
| drine | 내일 | tomorrow |
| marve | 봄 | spring |
| pess | 생각하다 | think |
| smow | 파다 | dig |
| gras | 이해하다 | understand |
| zerk | 요리하다 | cook |
| hule | 가득한 | full |
| spoy | 화난 | angry |
| yeel | 긴 | long |
| dizz | 추운 | cold |
| tately | 후에 | later |
| novely | 조용히 | quietly |
| vapsely | 빨리 | quickly |
| breaply | 갑자기 | suddenly |

vocabulary [19]. Therefore, L1 glosses were used in this study. Table 1 summarizes the target words and glosses used in this study.

**3.3.3. Eye-tracker and eye-tracking stimulus.** In this study, participants were asked to read the text from the eye-tracker screen (SMI-RED 500; SensoMotoric Instruments, Germany), which measures participants' eye movements using a 500-Hz infrared camera attached to the bottom of an eye-tracker monitor. The monitor was a 15-inch screen that looked the same as a normal computer screen.

The eye-tracking stimulus comprised an experimental text divided into 34 slides. The text was typed using 23-point Arial font while the target words and glosses at the bottom margin of the text were typed in 20-point Arial font. To increase the eye-tracking accuracy, we used two spaces between words and applied double-spaced lines. Both TA and NTA groups used the same eye-tracking stimuli.

**3.3.4. Vocabulary post-tests.** This study performed three kinds of vocabulary post-tests to evaluate the participants' learning of target words after they read the text, including (1) form recall test, (2) meaning recall test, and (3) meaning recognition test. First, the form recall test assessed the ability to recall the target word form (i.e., spelling) presented with glosses at the bottom of the text. The participants were allocated two minutes to write down the form of the 16 target words in the blanks of the test sheet. In case they did not remember the words, participants could leave blanks. See Appendix II in S1 File for form recall test items. Second, a mean recall test was administered to measure whether participants remembered the meanings of the target words. For example, participants had to write the meaning of the word next to the form of the target word (e.g., fusk) either in English (noise) or Korean (소음) when it was presented on the test paper. See Appendix III in S1 File for meaning recall test items. Third, the participants were allowed two minutes to complete a multiple-choice meaning recognition test in which they chose the meaning of the target word in Korean from five examples. The last of the five options was "I don't know" to prevent participants from randomly choosing answers to questions they did not know. For example, in the case of the target word "fusk," participants could choose the appropriate meaning among the five options: (1) 움직임 (motion), (2) 소음

(noise), (3) 연기 (smoke), (4) 가방 (bag), (5) 모르겠음 (I don't know). See Appendix IV in S1 File for meaning recognition test items.

In summary, the first test, the form recall test, aimed to evaluate how much the participant had acquired the form of target words, and the second and third tests, meaning recall and meaning recognition tests, were intended to assess the form-meaning association of target words. We used two different form-meaning association tests to examine whether test announcement can affect the mapping of form to meaning differently according to different tests. More specifically, the meaning recall test measures relatively stronger form-meaning association by asking to recall a meaning of a given target word, rather than the meaning recognition test that allows to choose the best answer among multiple-choices. These three vocabulary post-tests were performed in the same way as the immediate and delayed vocabulary tests.

**3.3.5. Reading comprehension test.** A reading comprehension test was used to measure how well the participants understood the content of the text by measuring both main idea and detailed information. The reading comprehension test featured 10 multiple-choice questions which tested participants' understanding of the text and three short-answer questions which measured the memory of key details of the text. These two kinds of questions were used, because short answer questions by eliciting constructed response can measure reading comprehension of the participants better than multiple-choice questions. In answering multiple-choice questions participants, on the other hand, can infer the correct answer from the given choices. They were not allowed to refer back to the reading text while answering the questions, because additional exposures to the target words by rereading the text may cause another variable in measuring participants' behavior.

In the reading comprehension test, some questions were related to the target words. For example, in order to answer a particular question asking which character did a specific behavior in which season, a participant needed to remember the form and meaning of a target word 'marve' meaning '봄' (*spring* in Korean) presented as an option out of four choices. For a reading comprehension test item in which 'marve' is an answer, see question 5 in Appendix I in S1 File. Furthermore, the reading comprehension test aimed to examine whether the TA group, who received prior notice of the vocabulary test, would exhibit increased attention to vocabulary learning and thus demonstrate a reduced understanding of the text than the NTA group. The test lasted four minutes and all questions and instructions were provided to the participants using L1 (Korean). See Appendix I in S1 File for all reading comprehension test items.

## 3.4. Procedure

The experimental procedure is illustrated in Fig 2. Particularly, the following order was followed: (1) text reading through an eye-tracker; (2) reading comprehension test; (3) immediate vocabulary post-tests (i.e., form recall test, meaning recall test, meaning recognition test); and (4) delayed vocabulary post-tests (i.e., one week later). First, the participants were tested individually in an eye-tracker lab. At the beginning of the experiment, participants were guided about the eye-tracker and several settings were implemented to ensure that participants maintained a comfortable posture and that eye-tracking was accurate. Specifically, the distance between the eye-tracking monitor and the participants was set to approximately 700 mm. Furthermore, a chin rest was used to minimize the participants' head movements, while a calibration procedure was performed to collect the correct gaze points. After setting up the eye-tracker, the participants were instructed on how to read the text. The participants were informed that they could read the entire text comprising 34 slides while pressing the space bar to navigate to the next slide. However, the participants received different notices for the

**Fig 2. Schematic description of the experimental procedure.**

upcoming vocabulary post-tests depending on their group. Particularly, the TA group was notified in advance that two kinds of test would be administered after reading (i.e., a "vocabulary test" and a "reading comprehension test"). In contrast, the NTA group did not receive any notice of the vocabulary test and was only informed about the "reading comprehension test." Therefore, the difference between the TA and NTA groups was whether prior notice was provided for the "vocabulary test." After the participants finished reading the text, they moved to another desk and completed the paper-based reading comprehension test and three kinds of vocabulary post-tests using a pencil. Vocabulary tests were conducted in the following order: form recall, meaning recall, and meaning recognition. This order was used because the form of the target word was presented in the meaning recall and meaning recognition tests. Thus, the participants could have learned the form of the target word by participating in the test, thus affecting the form recall test. Finally, after a week of delay, the participants returned to complete the same three vocabulary tests in the same order as the immediate vocabulary post-tests.

## 3.5. Scoring

All tests were scored using a binomial method that scored 1 point for a correct answer and 0 points for an incorrect answer. In the form recall test, only the correct spelling of the target word was considered the correct answer. Therefore, the answer was deemed incorrect even if only one letter was wrong. In the meaning recall test, an answer could be accepted as correct if the participant wrote a synonym instead of the same words presented in the gloss. For example, the answer was "noise," but "loud sound" was also accepted as the correct answer. Third, the meaning recognition test was multiple-choice where a participant received 0 points for choosing either the wrong option out of the four options or (5) "I do not know." Finally, the same scoring method as for the meaning recognition test was used to evaluate multiple-choice questions in the reading comprehension test. The three short-answer questions in the reading comprehension test were scored leniently, and students received 1 point if the meaning was correct despite minor spelling or grammatical errors.

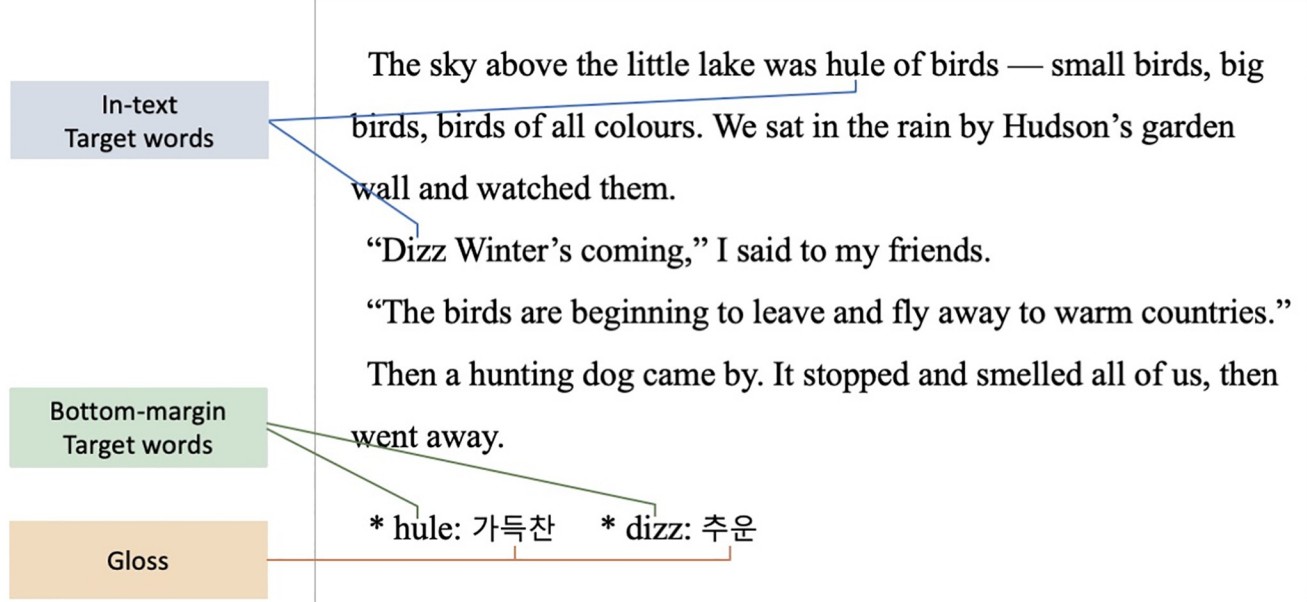

**Fig 3. Three ROIs in the eye-tracking stimuli.**

### 3.6. Preprocessing and analysis of eye-movement data

Eye-movement data were pre-processed using the default configuration of the commercial software package BeGaze, a package developed by SensoMotoric Instruments (Teltow, Germany). Next, the eye-movement data was visually checked for each participant and a drift correction was performed when necessary. Additionally, to further assess the quality of the eye-movement data, we used a high eye-tracking ratio cut-off point (90%), which led to eye-tracking data from 5 participants excluded [19, 23]. Tracking loss is defined as the 'the proportion of time that the eye tracker recorded point of gaze coordinates over the entire task' [24].

The eye-fixation data represent spatial and temporal data indicating where and for how long the participants fixed their gaze while reading a text. We set an ROI for each of three words: *in-text* target word, *bottom-margin* target word, and gloss, respectively (See Fig 3). Three measures were used to analyze these ROIs: (1) gaze duration, (2) second-pass reading time, and (3) total reading time. Gaze duration refers to the total time from when readers first fixed their eyes on the ROI, excluding the time they left the ROI and returned to gaze at it. Second-pass reading time refers to the sum of the fixation times from the second visit to the ROI. Finally, the total reading time refers to the sum of all fixation times in the ROI. Moreover, gaze duration is commonly associated with lower-level or automated vocabulary processing, while second-pass reading time and total reading time suggest enhanced strategic processing, such as memorizing vocabulary.

## 4. Results

### 4.1. Eye-tracking results

The eye-tracking results are reported in the order of the three ROIs (i.e., in-text target words, bottom-margin target words, and glosses). Consequently, three eye-tracking measures (i.e., gaze duration, second-pass reading time, and total reading time) were computed for these regions. The amount of time reading the experimental text varied among the learners.

**Table 2. Processing of target words and glosses (time in seconds; standard deviations in parenthesis).**

| Group | In-text target words | | | Bottom-margin target words | | | Glosses | | |
|---|---|---|---|---|---|---|---|---|---|
| | Gaze duration | Second pass reading time | Total reading time | Gaze duration | Second pass reading time | Total reading time | Gaze duration | Second pass reading time | Total reading time |
| TA | 1.98 (0.94) | 1.41 (1.03) | 3.39 (1.83) | 0.96 (0.47) | 0.73 (1.22) | 1.69 (1.51) | 0.43 (0.24) | 0.12 (0.19) | 0.55 (0.36) |
| NTA | 2.13 (1.24) | 1.41 (1.10) | 3.54 (2.24) | 0.67 (0.31) | 0.15 (0.16) | 0.81 (0.42) | 0.51 (0.39) | 0.19 (0.32) | 0.69 (0.63) |

Therefore, the eye-tracking data for the three ROIs were computed in terms of the proportion of time [25, 26]. For example, gaze duration for glosses = (raw gaze duration for glosses/total reading time of the entire experimental passage) × 100 (see Table 2).

**4.1.1. Processing of *in-text* target words.** The target words included in the text were referred to as "in-text target words." An independent samples *t*-test was performed to analyze whether the time used to process *in-text* target words differed significantly between the TA and NTA groups. Consequently, the result showed no significant difference between the TA and NTA groups in gaze duration ($t_{63} = 0.552$, $p = 0.583$, $d = 0.137$), second-pass reading time ($t_{63} = 0.001$, $p = 0.999$, $d < 0.001$), or total reading time ($t_{63} = 0.300$, $p = 0.765$, $d = 0.074$).

**4.1.2. Processing of *bottom-margin* target words.** An independent samples *t*-test was conducted to examine whether there was a significant difference in the time used to process *bottom-margin* target words between the TA and NTA groups. The findings indicated that the time to process *bottom-margin* target words was higher in TA than the NTA group for gaze duration ($t_{63} = 2.905$, $p = 0.005$, $d = 0.719$), second-pass reading time ($t_{63} = 2.745$, $p = 0.01$, $d = 0.676$), and total reading time ($t_{63} = 3.197$, $p = 0.003$, $d = 0.788$).

**4.1.3. Processing of glosses.** An independent samples *t*-test was run to see whether the time spent reading the glosses differed significantly between the TA and NTA groups. No significant differences were found between the TA and NTA groups for gaze duration ($t_{63} = 0.990$, $p = 0.327$, $d = 0.246$), second-pass reading time ($t_{63} = 1.043$, $p = 0.301$, $d = 0.258$), or total reading time ($t_{63} = 1.147$, $p = 0.257$, $d = 0.286$).

## 4.2. Vocabulary test results

**4.2.1. Form recall.** The results of the form recall test are illustrated in Table 3. These findings showed that the TA group received higher scores for both the immediate (*M* = 3.21) and delayed (*M* = 1.57) tests in contrast to the NTA group (immediate: *M* = 1.90, delayed: *M* = 0.81). A two-way mixed analysis of variance (ANOVA) was conducted to assess whether this difference was significant. The results showed a significant main effect for the "test announcement" ($F_{1,63} = 10.335$, $p = 0.002$, partial eta$^2$ = 0.141) and "period" ($F_{1,63} = 44.781$, $p < 0.001$, partial eta$^2$ = 0.415). However, no significant interaction was observed between "test announcement" and "period" ($F_{1,63} = 1.769$, $p = 0.188$, partial eta$^2$ = 0.027).

Based on the significant main effect of the "test announcement," an independent samples *t*-test was run to compare the results of the form recall test for each of the immediate and delayed tests between the TA and NTA groups. This study found that the TA group recalled

**Table 3. Results from three vocabulary tests (standard derivations in parenthesis).**

| Group | Form recall | | Meaning recall | | Meaning recognition | |
|---|---|---|---|---|---|---|
| | immediate | delayed | immediate | delayed | immediate | delayed |
| TA | 3.21 (1.91) | 1.57 (1.50) | 8.6 (3.56) | 6.24 (3.25) | 13.06 (2.69) | 12.18 (3.49) |
| NTA | 1.90 (1.59) | 0.81 (0.96) | 7.18 (2.75) | 5.15 (2.85) | 11.3 (2.56) | 10.84 (2.64) |

more target word forms than the NTA group in the immediate form recall test, and the difference was statistically significant ($t_{63} = 2.982$, $p = 0.004$, $d = 0.74$). Similarly, the TA group recalled more target words than the NTA group during the delayed form recall test ($t_{63} = 2.430$, $p = 0.018$, $d = 0.61$). Thus, the TA group scored higher than the NTA group on both the immediate and delayed tests.

According to the significant main effect of "period," an independent samples $t$-test was conducted to compare the TA and NTA groups in immediate and delayed tests. In the TA group, the immediate form recall test scored higher than the delayed test, and the difference was statistically significant ($t_{32} = 5.959$, $p < 0.001$, $d = 0.74$). Furthermore, the NTA group scored higher on the immediate test than on the delayed test, and the difference was statistically significant ($t_{31} = 3.617$, $p = 0.001$, $d = 0.83$). Thus, both TA and NTA groups scored higher on the immediate form recall test in contrast to the delayed form recall test.

**4.2.2. Meaning recall.**    The results of the meaning recall test indicated that the TA group scored slightly higher for both the immediate and delayed ($M = 8.6$, $6.24$, respectively) tests as opposed to the NTA group ($M = 7.18$, $5.15$). However, a two-way mixed ANOVA revealed that the effect of the "test announcement" did not yield a significant difference between the TA and NTA groups ($F_{1,63} = 3.081$, $p = 0.084$, partial eta$^2 = 0.047$). However, the main effect of the "period" was statistically significant ($F_{1,63} = 60.580$, $p < 0.001$, partial eta$^2 = 0.490$), whereas the interaction between "test announcement" and "period" was not significant ($F_{1,63} = 0.407$, $p = 0.526$, partial eta$^2 = 0.006$).

According to the significant main effect of "period," an independent samples $t$-test was performed to compare the immediate and delayed meaning recall test of TA and NTA groups. Consequently, both the TA ($t_{32} = 5.706$, $p < 0.001$, $d = 0.70$) and NTA groups ($t_{31} = 5.314$, $p < 0.001$, $d = 0.72$) exhibited significantly higher scores on the immediate test than on the delayed test.

**4.2.3. Meaning recognition.**    The results of the meaning recognition test showed that the TA group scored the highest among the three vocabulary tests (immediate: $M = 13.06$, delayed: $M = 12.18$) in contrast to the NTA group (immediate: $M = 11.3$, delayed: $M = 10.84$). A two-way mixed ANOVA revealed that the main effect of "test announcement" was not significant ($F_{1,63} = 3.358$, $p = 0.072$, partial eta$^2 = 0.051$), whereas the main effect of "period" was significant ($F_{1,63} = 16.716$, $p < 0.001$, partial eta$^2 = 0.210$). Furthermore, the interaction between "test announcement" and "period" was not significant ($F_{1,63} = 0.199$, $p = 0.657$, partial eta$^2 = 0.003$).

Although the effect of "test announcement" showed no significance, the main effect of "period" was significant. Accordingly, an independent samples $t$-test was conducted to compare the scores on the immediate and delayed meaning recognition tests of the TA and NTA groups. Consequently, both the TA group ($t_{32} = 2.680$, $p = 0.012$, $d = 0.45$) and the NTA group ($t_{31} = 3.085$, $p = 0.004$, $d = 0.42$) scored significantly higher on the immediate test than on the delayed test.

## 4.3. Reading comprehension test results

To determine whether vocabulary test announcements negatively influenced reading comprehension, the reading comprehension test scores were analyzed. The results showed similar scores for the TA ($M = 10.55$, $SD = 1.54$) and NTA groups ($M = 10.94$, $SD = 1.22$). Moreover, an independent samples $t$-test showed no statistically significant difference between groups ($t_{63} = 1.135$, $p = 0.261$, $d = 0.28$).

## 5. Discussion

This study investigated how the announcement of a test influences the attention, vocabulary acquisition, and reading comprehension of L2 learners when they read an English text that

uses glosses. In doing so, the current study used both online (eye-tracking) and offline (paper-based test) methods. Consequently, this study found that test announcements affected participants' attention as well as the aspect of vocabulary knowledge they acquired while reading English text with glosses. In addition, test announcements exhibited no negative effect on reading comprehension; that is, this study did not detect trade-off effects, which would normally indicate a negative effect on reading comprehension when more attention is allocated to vocabulary uptake in response to the announcement of the test. These findings are discussed in the same sequence as the research questions.

The first research question proposed: *What are the effects of a test announcement on learners' attention*? As mentioned earlier, the current study used eye-tracking to investigate participants' attention. The eye-tracking results showed that there was a significant difference between TA and NTA groups only in processing *bottom-margin* target words. In other words, the TA group spent significantly more time processing *bottom-margin* target words than the NTA group for all three eye-tracking measures (i.e., gaze duration, second-pass reading time, and total reading time).

Notably, in contrast to reading *bottom-margin* target words, the test announcement did not yield a significant effect on reading *in-text* target words. This result suggests that test announcements could encourage the increased use of cognitive resources among learners for processing and memory of *bottom-margin* target words than for *in-text* target words. In the context of incidental or intentional vocabulary acquisition, *in-text* target words could be processed as part of the course of learners' understanding of the text and thus be regarded as incidental. In contrast, *bottom-margin* target words were often acquired more explicitly and intentionally. Thus, further research using interviews or think-aloud techniques could explore this possibility.

The TA and NTA groups did not show significant differences in the reading time of *the bottom-margin* glosses. One possible explanation for this result is that the glosses used in the study were presented in learners' L1 (Korean). Therefore, the words did not require extensive processing effort as the information had been established using L1 speakers' mental lexicon. Furthermore, the L1 glosses did not constitute the information to be newly acquired. Thus, the limited importance associated with learning these words could explain the lack of significant difference between the two groups in glosses' reading times.

The second question of this study was: *What are the effects of a test announcement on learners' vocabulary acquisition (i.e., form acquisition and form-meaning association)*? Upon analyzing the vocabulary tests, the findings showed that test announcements had a positive effect on form acquisition (i.e., form recall test), but not on form-meaning association (i.e., meaning recall test and meaning recognition test). These findings were partially consistent with previous studies which showed that test announcements did not have a significant effect on form-meaning mapping [3, 5]. Unlike the findings of the current study, the lack of significant difference between the TA and NTA groups might have occurred because Ko (1995) [3] did not examine form acquisition of vocabulary. Peters (2007) [5] used a form recognition test as opposed to a form recall test, which was used in the present study. Furthermore, the difference between these previous and current results might have occurred because recall and recognition (i.e., of forms) involved different cognitive efforts. In other words, recognition test required choosing an answer among many options and allowed a chance effect, thus, involved relatively fewer cognitive efforts than recall test, where no clue was provided. It is possible that the difference between TA and NTA groups will be greater for tests in which relatively more cognitive efforts (i.e., recall) were required than for tests in which relatively fewer cognitive efforts (i.e., recognition) were needed. Moreover, most previous research failed to obtain positive effects of test announcements on vocabulary. However, this study indicated that vocabulary test

announcements had a positive effect on the acquisition of the form of target words, which supported the findings by Peters et al. (2009) [6].

The current results revealed that the effect of vocabulary test announcements could vary according to the kind of vocabulary test: form recall vs. meaning recall. Furthermore, this study observed that learners needed additional cognitive efforts more for learning form-meaning associations than for learning word form. Thus, test announcements could help students to encode forms of new words. However, additional cognitive resources will be required to map forms to meanings of encoded words, or to consolidate form-meaning associations of the encoded words.

The final research question was: *What are the effects of a test announcement on reading comprehension*? Consistent with previous research, this study showed that test announcements did not produce a significant effect on reading comprehension test scores [3, 6]. The trade-off effect that states that prereading test announcements allow learners to increase focus on the target words while reducing attention on reading comprehension was not observed in this study. Therefore, even though test announcements were likely to increase the attention devoted to new words, the diversion of this attention did not negatively affect learners' reading comprehension.

The findings of this study underscored the implications of test announcements prior to reading a glossed text in vocabulary acquisition. However, the results should be interpreted with caution, considering that this study had several limitations.

First, we provided glosses in the participants' L1 (Korean). According to Kang et al. (2020) [19], L1 glosses in Korean were more effective than L2 glosses in English for vocabulary learning (Hedge's g = 0.27, p < 0.01). Therefore, the findings on learners' attention and reading behaviors examined with L1 glosses might have been different if L2 glosses had been used.

Second, the announcement of the vocabulary test did not include specific information about the kind of test. Thus, details indicating the kind of vocabulary test could have influenced the aspect of vocabulary knowledge that the participants learned. Thus, future research should investigate whether awareness of the kind of vocabulary test (i.e., form recall, meaning recall, and meaning recognition) affects learners' attention and the aspect of vocabulary knowledge learned. Learning new words while reading is basically an input-based learning activity. Therefore, future studies may need to incorporate a form recognition test as a measure of lexical gains. As a reviewer suggested, further research should focus on L2 learners' depth of processing and the mechanisms leading to differential learning outcomes of word form and meaning to see the effect of test announcement.

Third, the current study used ordinary text that had low lexical density and required no special background knowledge. Thus, the results might have varied if different kinds of texts were used. Considering the effect of background and lexical knowledge on reading comprehension [1], this experiment should be repeated using a different kind of text from the one used in this study.

Finally, this study did not examine the possibility of a correlation between learner variables such as English proficiency, working memory, and test announcements. Further research involving participants with different proficiency levels or working memory than those in the present study needs to be conducted to observe the rigorous effects of test announcements on vocabulary acquisition.

In conclusion, this study showed that test announcements affected the distribution of time, which indicates where and how long learners paid attention to reading glossed texts. Additionally, test announcements seemingly affected the aspect of vocabulary information learned when learners read texts accompanied by glosses. Furthermore, that test announcement was only useful for learning how to form new words, but not for consolidating form-meaning

mapping. Thus, this observation could suggest that the latter requires more cognitive effort or rehearsal in contrast to the former. Moreover, the test announcement did not reduce reading comprehension. Test announcements could serve as a useful means of vocabulary learning without sacrificing learners' reading comprehension as well as be applied to class guidance and action planning in L2 classrooms. In addition, the use of eye-tracking in the present study to investigate the effect of glosses and test announcements on information about learners' attention in quantitative ways had research implications. Despite the valuable contributions, further research should be conducted to verify the results and mitigate the limitations of this current study.

## Supporting information

**S1 File.**
(DOCX)

## Author Contributions

**Methodology:** Hayoung Kim.

**Supervision:** Sungmook Choi.

**Writing – original draft:** Soo-Ok Kweon.

**Writing – review & editing:** Soo-Ok Kweon.

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
