## [Decision Letter · Decision Letter 0]

18 Jul 2022

PONE-D-22-15213Effects of vocabulary test announcement prior to reading a glossed text on reading behaviors and vocabulary acquisition: An eye movement studyPLOS ONE

Dear Dr. Kweon,

Thank you for submitting your manuscript to PLOS ONE. After careful consideration, we feel that it has merit but does not fully meet PLOS ONE’s publication criteria as it currently stands. Therefore, we invite you to submit a revised version of the manuscript that addresses the points raised during the review process.

 All of the reviewers and I see the value of your research and recognize the contribution it can make to the field. However, Reviewers 1 and 3 highlight some serious concerns, which I share. Some of the reviewers concerns are provided in supplementary files. Please carefully consider all of the points that the reviews have made, both within the PLOS ONE system and in the uploaded documents. All concerns should be addressed in the resubmission of your manuscript. 

We look forward to receiving your revised manuscript.

Kind regards,

Kathy Conklin, PhD

Academic Editor

PLOS ONE

Journal Requirements:

3. Please change "female” or "male" to "woman” or "man" as appropriate, when used as a noun (see for instance https://apastyle.apa.org/style-grammar-guidelines/bias-free-language/gender).

Reviewers' comments:

Reviewer's Responses to Questions

**Comments to the Author**

1. Is the manuscript technically sound, and do the data support the conclusions?

Reviewer #1: Yes

Reviewer #2: Yes

Reviewer #3: Yes

2. Has the statistical analysis been performed appropriately and rigorously? 

Reviewer #1: Yes

Reviewer #2: Yes

Reviewer #3: Yes

3. Have the authors made all data underlying the findings in their manuscript fully available?

Reviewer #1: Yes

Reviewer #2: Yes

Reviewer #3: Yes

4. Is the manuscript presented in an intelligible fashion and written in standard English?

Reviewer #1: Yes

Reviewer #2: Yes

Reviewer #3: Yes

5. Review Comments to the Author

Reviewer #1: This is an interesting study that explored how test announcement prior to reading would moderate the impact of glossing on learning of form and meaning of target words. The authors employed eye-tracking technology to measure the amount of attention driven to the in-text as well as text-margin target words, and their glosses. The learning was assessed in terms of form and meaning recognition and recall abilities. The results revealed that test announcement did not have significant impact on reading comprehension, but promoted attention to the text-margin target words and form recall scores.

While this study certainly points out a meaningful gap in the literature and provides pedagogical insights on how to better utilize glosses in L2 reading instruction, there are issues that should be addressed to be considered for publication in the journal. More detailed comments are provided below.

p. 3

Before explaining incidental and intentional learning here, I would recommend authors to briefly review and clarify the relationship between implicit/explicit language learning and incidental/intentional language learning.

Plus, it is not that we can simply equate incidental learning as implicit learning. For instance, we do sometimes look up in a dictionary to search a meaning of an unknown word, while reading for pleasure with no liabilities.

“However, during incidental vocabulary learning, frequent exposure to language input may take a long time to be connected to the acquisition of the vocabulary, the wrong meaning of words may be learned, if the learner is not provided with correct cues to help retrieve the meanings of unknown words from context.”

>> citation please.

p. 4

This section reviews only a few glossing studies in the context of incidental learning.

- Hulstijn et al., 1996

- Ko, 2012

- Yanagisawa et al., 2020 (meta-analysis)

- Kang et al., 2020

- Choi, 2016

Why is that? There is an extensive amount of research on the impact of glossing on incidental vocabulary learning, and I'm curious why the authors reviewed only these studies..

Plus, given that one of the purposes of this study was to prove if glossing indeed promotes attention to the glossed words (as well as the glosses), they would want to review more eye-tracking studies on glossing (that is, under the no-test-announcement condition, not just Choi (2016) and Kang t al. (2020).

In the first paragraph of 2.2, Please provide more detailed explanation on the cognitive/ pedagogical mechanism in which glosses can promote incidental vocabulary learning.

p. 5

2.3

This section reads like a patchwork, loosely listing the studies without a clear logical sequence. I would recommend the authors to make this section more organic and logical, demonstrating how they synthesize the methodological aspects and the findings of the reviewed studies for readers.

"Many studies have quantified the glossing effect in incidental vocabulary learning, whereas relatively fewer studies seem to be done on the effect of glosses in the context of intentional vocabulary learning (cf. Ko, 1995; Montero Perez et al., 2018; Peters, 2007; Peters et al., 2009; Zuo, 2021)."

>> Why do the authors think is the case?

p. 6-7

Another a bit awkward transition.. The authors should make this section more condensed, synthesized, and logical. (e.g., Further, Furthermore, Afterwards, ….) This issue pertains to the manuscript as a whole, and in particular this section (pp. 6-7).

p. 9

4.3.1.

This is good information on the reading text, but please additionally add the lexical coverage (2K and 3K?) as well as readability index.

"The Turkish names of the characters were changed to English names to make it easier for participants to read. For example, ‘Hikmet’ was replaced by ‘Hudson’, and ‘Halil’ was replaced by ‘Morgan’."

>> Please add explanation for this with proper citations

4.3.2.

"The reason that pseudowords were used as the target words is that the subject's prior vocabulary knowledge may affect the experimental results."

>> Add citation please.

More importantly, why were these particular words chosen? Please provide the conditions. For instance, why didn’t the authors control the part of speech of the target words? What about their task-essentialness (that is, were they important in successful reading comprehension)?

p. 10

4.3.4.

Why not form recognition? Given this was an input-based learning activity, I'd like to know why the authors did not measure form recognition.

In the form recall test, what prompt was given? Were the participants provided with meanings? Or just 16 blanks? Or were the blanks provided within meaningful context? Please provide a sample item to help readers' understanding.

4.3.5

Please add more detailed information on the RC questions, not just their formats. For example, what dimensions of RC was measured? Main idea/ detailed information/ true-not true informaation/ finding synonyms/ etc. Also, what was the rationale for including two different types of RC questions? More importantly, were they allowed to refer back to the reading text while answering the questions? Or they had to rely on their memory? Why?

What was the reliability? I would recommend the authors to share the text and the RC questions as appendix or supplementary item.

p. 11

4.5.

Without detailed information on the short-answer RC questions, this scoring method seems not very useful.

p. 12

4.6.

Please add information on how the raw eye-movement data were trimmed before conducting statistical analysis. For example, how were the outliers treated/ filtered out? Also, what statistical anslysis was used? Please provide detailed information on how the pre-processed eye-movement data were analyzed. Also, how were the different reading times across participants handled? This section lack too much important information that is necessary to replicate this study, so I would suggest extensive revision here.

5.1.1.

d > 0.001 ??? >> d < 0.001

p. 13

“An independent-samples t-test was performed to analyze whether the time…”

>> This exact wording is too repetitive.

Results >> The results

p. 15

The 2nd and the 3rd paragraphs are very repetitive and redundant. The authors would want to focus on interpreting the results in this section.

Plus, there are studies that used eye-tracking technology to examine if glossing works.

Were the target words necessary for comprehension? The authors would want to consider the task-essentialness of the target words. That is, if the target words didn’t matter (especially adjectives or adverbs), they could have felt only little need (Laufer and Hulstijn’s Involvement Load Hypothesis) to know the meaning of the target words.

p. 16

"The difference between these results and ours might have occurred because recall and recognition (i.e., of forms) involve different cognitive efforts."

>> Please elaborate more on this.

"Our results reveal that vocabulary test announcement can affect which type of vocabulary knowledge can be acquired."

>> I'm not sure if I can understand this part. The authors may want to revise it.

"That is, test announcement can help the students to encode forms of new words, but that additional cognitive resources will be required to map forms to meanings of encoded words, or to consolidate form-meaning association of the encoded words."

>> Glossing in and of itself, promotes learning of word meanings. Previous studies overall showed that glossing promotes learning of word meanings, rather than forms. To be more specific, when glossing is not provided, readers tend to inspect the text more intensively to infer the meaning of the unknown word, especially if the word is important in comprehension. When glossing is provided, they do not feel the need to search for the meaning, as the gloss is right there, and hence have no or even negative impact on learning word forms. Perhaps additional demands induced by the TA could have compensated this relatively weak/inhibitory impact of glossing on learning of word forms. The authors should review more studies on the impact of glossing on learning of word forms, and not to mention, Laufer and Hulstijn's Involvement Load Theory. w

"We presented fruitful findings on and implications of test announcement prior to reading a glossed text in vocabulary acquisition."

>> The authors would want to revise this sentence, especially the word choice "Fruitful."

"Information about the type of vocabulary test might have changed the type of vocabulary that the participants learned."

>> I am not very suer if the word "type" is recommendable. The authors would want to reconsider this term throughout the manuscript to enhance clarity.

The authors would want to use the past tense when reporting what they DID or FOUND in this study throughout the methodology, results, and discussion sections.

Reviewer #2: The researchers were able to address the research problem and discuss aspects of research from several parties. This was clear from the several procedures used to verify the glossing of immediate knowledge and delayed ones. Where the research problem emerged and an attempt to show the research gap in this research. The researchers proceeded scientifically in presenting the research ideas in all parts of the research. The researchers also used an academic language free of linguistic errors. The language used in presenting the previous related studies was relatively accurate and complete to the extent that it satisfies what is required in presenting opinions, facts, and previous results.

Reviewer #3: Comments on “Effects of vocabulary test announcement prior to reading a glossed text on reading behaviors and vocabulary acquisition: An eye movement study”

Some snapshots in the following cannot be displayed. Please refer to the attached file.

The paper is in general well written. The research was carefully designed and the use of eye-tracking method is innovative. There are a number of issues, however, that need to be considered by the researchers.

1. Concerning form / meaning distinction

It is good that the researchers, based on previous studies, went a step further to measure the effect of test announcement on form recall and meaning recall. The researchers, however, did no review literature about the form / meaning distinction in L2 vocabulary learning. Since form / meaning are key constructs in the research, there need to be more discussions about them in the literature review.

2. In the research, there are three ROIs. What are the reasons of setting three ROIs for each target word? Please state the rationale.

Isn’t the bottom margin-target word a part of the gloss?

3. Why are there F2 words and F4 words? The frequency of the target words is not a variable in this research.

4. What are the purposes of carrying out a meaning recall test and a meaning recognition test? Please justify this.

5. Was the form recall test more like a memory test instead of a vocabulary learning test?

6. The scoring method for the meaning recall test can be improved since it seemed to be more flexible compared with the form recall test. The researchers are encouraged to try: 0 – incorrect; 1 – partially correct; 2 – correct.

7. Another issue is about the reading comprehension questions. Are they related to the target words? Explanations are needed in the manuscript.

8. Are the target words important for the comprehension of the text? This is to be explained in the research design.

9. Test announcement is one step to initiate L2 learners’ noticing of the target word in reading. Whether the words are retained or not depends largely on the learner’s subsequent processing of the words, esp. depth of processing. As the researchers reviewed in the manuscript, a number of studies have been done on the effect of test announcement on L2 vocabulary learning, it is suggested that in future research the researchers focus more on processing depth and the mechanisms leading to differential learning outcomes of word form / meaning.

6. PLOS authors have the option to publish the peer review history of their article (what does this mean?). If published, this will include your full peer review and any attached files.

Reviewer #1: No

Reviewer #2: **Yes: **ALHARBI BADER

Reviewer #3: No

---

## [Author Response · Author response to Decision Letter 0]

27 Dec 2022

Reviewer 1

This is an interesting study that explored how test announcement prior to reading would moderate the impact of glossing on learning of form and meaning of target words. The authors employed eye-tracking technology to measure the amount of attention driven to the in-text as well as text-margin target words, and their glosses. The learning was assessed in terms of form and meaning recognition and recall abilities. The results revealed that test announcement did not have significant impact on reading comprehension, but promoted attention to the text-margin target words and form recall scores.

While this study certainly points out a meaningful gap in the literature and provides pedagogical insights on how to better utilize glosses in L2 reading instruction, there are issues that should be addressed to be considered for publication in the journal. More detailed comments are provided below.

p. 3 

Before explaining incidental and intentional learning here, I would recommend authors to briefly review and clarify the relationship between implicit/explicit language learning and incidental/intentional language learning. 

Plus, it is not that we can simply equate incidental learning as implicit learning. For instance, we do sometimes look up in a dictionary to search a meaning of an unknown word, while reading for pleasure with no liabilities. 

Response: 

Yes. We totally agree with your comment: “it is not that we can simply equate incidental learning as implicit learning” as the reviewer mentioned. Laufer and Hulstijn (2001) also argued that researchers should not be confused with the notions of incidental and intentional learning with the notions of implicit and explicit learning, as follows: 

Although implicit learning can be incidental only (i.e. without learners' awareness of an upcoming retention test, or without learners' deliberate decision to commit information to memory), explicit learning can occur both intentionally and incidentally. The relevance of the above distinctions to L2 vocabulary learning is not difficult to see. Since linking word form to word meaning is an explicit learning activity requiring attention on the part of the learner, vocabulary can therefore be learnt intentionally as well as incidentally.” (Laufer and Hulstijn, 2001, p.11). 

Our focus is to explore how a vocabulary test announcement prior to reading a glossed text influences the reading behaviors and subsequent vocabulary acquisition of L2 learners. Thus, we compared two groups of participants: a vocabulary test announcement (TA) or a no test announcement (NTA) group. Both TA (intentional) and NTA (incidental) conditions involve explicit learning. The process of vocabulary acquisition using a glossed text is explicit learning, because participants should pay attention to form-meaning connection for an upcoming vocabulary test. The notion of implicit learning is not considered or discussed in our study, because it is irrelevant for the vocabulary acquisition behaviors designed in our paper. Rather, we are interested in incidental and intentional vocabulary acquisition, both of which are explicit learning. Therefore, to “review and clarify the relationship between implicit/explicit language learning and incidental/intentional language learning” seem to go beyond the scope of our paper. However, we do appreciate the reviewer’s comment by which we had a chance to think about the relationship between different cognitive views on the representation and attainment of L2 vocabulary knowledge. 

Additionally, the definition of incidental and intentional vocabulary learning varies in the field. According to Hulstijn’s classification (Hulstijin, 1992, 2001), incidental vocabulary learning occurs through contextualized activities, in which the acquisition of vocabulary items occurs incidentally as an outcome of the reading, listening, speaking, or writing activity (Hulstijn, 2001). However, intentional vocabulary acquisition occurs through decontextualized activities, in which intentional vocabulary learning is defined as completing activities intentionally designed for word learning, such as crossword puzzles, word cards, or word lists with synonyms and antonyms (Webb, 2020) The current research adopts the classification of incidental and intentional vocabulary learning based on (de)contextualization. Thus, the distinction between the notions of implicit and explicit learning go beyond the scope of this paper. 

“However, during incidental vocabulary learning, frequent exposure to

language input may take a long time to be connected to the acquisition of the vocabulary, the

wrong meaning of words may be learned, if the learner is not provided with correct cues to

help retrieve the meanings of unknown words from context.”

citation please.

Response: 

We added citation and revised the above passage as follows

“However, since incidental vocabulary learning requires frequent exposure to language input, it is not easy for learners to be exposed to the same vocabulary repeatedly while reading books or watching TV, and it may take a long time to succeed in vocabulary acquisition. Moreover, the wrong meaning of words could be learned, if learners are not provided with correct cues that help them retrieve the meanings of unknown words from context (Hulstijn, 1992; Nation, 2001).”

p. 4

This section reviews only a few glossing studies in the context of incidental learning. 

- Hulstijn et al., 1996

- Ko, 2012

- Yanagisawa et al., 2020 (meta-analysis)

- Kang et al., 2020

- Choi, 2016

Why is that? There is an extensive amount of research on the impact of glossing on incidental vocabulary learning, and I'm curious why the authors reviewed only these studies..

Response: 

Though many studies have been done on the effect of glossing on incidental vocabulary learning, there are limitations in reviewing them due to the small number of samples or a different research design from this study. Moreover, most of the glossing studies have already reported the positive effect of glossing on incidental vocabulary learning. As shown in recent meta-analysis of 42 glossing studies (Yanagisawa, 2020), more vocabulary was acquired from glossed reading than reading without glosses. Thus, we considered it appropriate to review only a few major studies published in top journals in the field.

Plus, given that one of the purposes of this study was to prove if glossing indeed promotes attention to the glossed words (as well as the glosses), they would want to review more eye-tracking studies on glossing (that is, under the no-test-announcement condition, not just Choi (2016) and Kang t al. (2020).

Response: 

Please understand that the only eye-tracking study investigating the effect of glosses on attention was that of Kang (2020). There is another eye-tracking study using glossed text by Warren et al. (2018), but they examined the effects of gloss types (pictorial, textual, and multimodal glosses) and they did not have a no-gloss group. Thus, it could not confirm whether glossing indeed promotes learners’ attention to the target words and glosses.

In the first paragraph of 2.2, Please provide more detailed explanation on the cognitive/ pedagogical mechanism in which glosses can promote incidental vocabulary learning. 

Response: 

Thank you for your valuable comments. We added the following part in 2.2.

“… Learners can gain a new vocabulary word by simply reading L2 text but providing gloss can further promote the vocabulary learning process. This is because gloss may allow learners opportunities to notice the lexical items in the reading materials and also to figure out the exact meaning of the new words, eliminating the risk of inferring the wrong meaning.”

p. 5

2.3

This section reads like a patchwork, loosely listing the studies without a clear logical sequence. I would recommend the authors to make this section more organic and logical, demonstrating how they synthesize the methodological aspects and the findings of the reviewed studies for readers.

.

Response: 

Thank you for an important suggestion. We significantly revised 2.3.

Many studies have quantified the glossing effect in incidental vocabulary learning, whereas

relatively fewer studies seem to be done on the effect of glosses in the context of intentional

vocabulary learning (cf. Ko, 1995; Montero Perez et al., 2018; Peters, 2007; Peters et al., 2009; Zuo, 2021).

Why do the authors think is the case?

Response: 

We have found very few numbers of studies on the effect of glossing on intentional vocabulary learning. Based on the results that glossing is effective for ‘incidental vocabulary learning’, further studies have been conducted on the effects of glossing according to its language (L1/L2) or various type (gloss with picture, audio, etc.) in the context of ‘incidental vocabulary learning’. However, there seems to be little interest in further treatment such as intentional vocabulary learning. We assumed that if glosses were found to have no effect in the context of incidental vocabulary learning, researchers might have turned their attention to the effectiveness of glossing in intentional vocabulary learning.

p. 6-7

Another a bit awkward transition.. The authors should make this section more condensed, synthesized, and logical. (e.g., Further, Furthermore, Afterwards, ….) This issue pertains to the manuscript as a whole, and in particular this section (pp. 6-7).

Response: 

We made a significant revision following your comments. Please look at pp. 5-7 of the revised manuscript. 

p. 9

4.3.1.

This is good information on the reading text, but please additionally add the lexical coverage (2K and 3K?) as well as readability index.

Response:

The text used in this study is level 1 of Oxford graded readers, which range from starter to level 7. Since graded readers are controlled texts using high frequency words for second language learners, we consider it not necessary to evaluate the lexical coverage of this text. However, we found the readability of the text through Flesch-Kincaid Grade Level, and the result was level 2.1. It means that native speakers with two years of formal education can read the text without difficulty. The readability index has already been provided in the paper, but we have rewritten it with the exact figures as below in revised manuscript. 

 “… According to the Flesch-Kincaid Grade Level, the readability of the text was Level 2.1, indicating that a native speaker with two years of formal education can read the text without much difficulty.”

The Turkish names of the characters were changed to English names to make it easier for participants to read. For example, ‘Hikmet’ was replaced by ‘Hudson’, and ‘Halil’ was replaced by ‘Morgan’.

Please add explanation for this with proper citations

Response: 

Unfortunately, to the best of our knowledge, we are not aware of any previous research that can be cited with respect to this kind of change of target words. 

We added the following paragraph in the revised manuscript. 

“Since this study is about vocabulary learning that occurs during text reading, we wanted to avoid presenting new or unknown words as much as possible other than target words in the text. Turkish names such as “Hikmet”, “Halil”, and “Tekin” would be less familiar to EFL Korean learners than English names and may be considered new words by them. Therefore, we modified the Turkish names to the English names to prevent participants from recognizing those words as target words, so that the purpose of this study could be more clearly examined.”

4.3.2.

The reason that pseudowords were used as the target words is that the subject's prior vocabulary knowledge may affect the experimental results.

Add citation please. 

Response: 

We added citation of Webb (2007), Choi (2016) and revised the part as below.

“This study used pseudowords as target words because the subjects’ prior vocabulary knowledge might have affected the experimental results (Webb, 2007; Choi, 2016).”

More importantly, why were these particular words chosen? Please provide the conditions. For instance, why didn’t the authors control the part of speech of the target words? What about their task-essentialness (that is, were they important in successful reading comprehension)?

Response: 

Vocabulary will play an important role in reading comprehension, because of the close relationship between the two. However, it is the case that many variables such as decoding speed, grammatical knowledge, background knowledge, language proficiency, etc. will also influence reading comprehension. Thus, it is difficult to answer as of now to what extent the target words affected reading comprehension. 

We added the following in the revised manuscript. 

“We chose F2 or F4 words as target words from the reading text, following Choi (2016). It was extremely difficult to find target F2 or F4 words that belong to the same part of speech, because glosses are used for words of different part of speech in the text to which L2 learners are exposed.”

p. 10

4.3.4.

Why not form recognition? Given this was an input-based learning activity, I'd like to know why the authors did not measure form recognition.

Response: 

Based on your comment, we added the following in the revised manuscript on pp. 20. 

“Learning new words while reading is basically an input-based learning activity. Therefore, future studies may need to incorporate a form recognition test as a measure of lexical gains.”

In the form recall test, what prompt was given? Were the participants provided with meanings? Or just 16 blanks? Or were the blanks provided within meaningful context? Please provide a sample item to help readers' understanding.

Response: 

No prompt was provided, as in previous studies (e.g., Choi, 2016). The participants were provided with just 16 blanks without context. We added the form recall test items in Appendix II in the revised manuscript. 

Choi, S. (2016). Effects of L1 and L2 glosses on incidental vocabulary acquisition and lexical representations. Learning and Individual Differences, 45, 137-143.

4.3.5

Please add more detailed information on the RC questions, not just their formats. For example, what dimensions of RC was measured? Main idea/ detailed information/ true-not true informaation/ finding synonyms/ etc. Also, what was the rationale for including two different types of RC questions? More importantly, were they allowed to refer back to the reading text while answering the questions? Or they had to rely on their memory? Why?

Response: 

We added the following in the revised manuscript. 

“A reading comprehension test was used to measure how well the participants understood the content of the text by measuring both main idea and detailed information. The reading comprehension test featured 10 multiple-choice questions which tested participants’ understanding of the text and three short-answer questions which measured the memory of key details of the text. The two types of questions were used, because short answer questions by eliciting constructed response can measure reading comprehension of the participants better than multiple-choice questions. In answering multiple-choice questions participants, on the other hand, can infer the correct answer from the given choices. They were not allowed to refer back to the reading text while answering the questions, because additional exposures to the target words by rereading the text may cause another variable in measuring participants’ behavior.”

 What was the reliability? I would recommend the authors to share the text and the RC questions as appendix or supplementary item. 

Response: 

Reliability was not calculated for RC test. The reason is that the test included not only multiple-choice items, but also constructed response items. As is mentioned in the manuscript, the constructed responses were elicited in short-answer questions to measure memory and understanding of target words. 

p. 11

4.5.

Without detailed information on the short-answer RC questions, this scoring method seems not very useful.

Response: 

We revised “3.3.5. Reading comprehension test” as following in the revised manuscript.

“A reading comprehension test was used to measure how well the participants understood the content of the text by measuring both main idea and detailed information. The reading comprehension test featured 10 multiple-choice questions which tested participants’ understanding of the text and 3 short-answer questions which measured the memory of key details of the text. These two kinds of questions were used, because short answer questions by eliciting constructed response can measure reading comprehension of the participants better than multiple-choice questions. In answering multiple-choice questions, participants, on the other hand, can infer the correct answer from the given choices. They were not allowed to refer back to the reading text while answering the questions, because additional exposures to the target words by rereading the text may cause another variable in measuring participants’ behavior.”

p. 12

4.6.

Please add information on how the raw eye-movement data were trimmed before conducting statistical analysis. For example, how were the outliers treated/ filtered out? Also, what statistical anslysis was used? Please provide detailed information on how the pre-processed eye-movement data were analyzed. Also, how were the different reading times across participants handled? This section lack too much important information that is necessary to replicate this study, so I would suggest extensive revision here. 

Response: 

Following your comment, we added the following in the revised manuscript. 

“Eye-movement data were pre-processed using the default configuration of the commercial software package BeGaze, a package developed by SensoMotoric Instruments (Teltow, Germany). Next, the eye-movement data was visually checked for each participant and a drift correction was performed when necessary. Additionally, to further assess the quality of the eye-movement data, we used a high eye-tracking ratio cut-off point (90%), which led to eye-tracking data from 5 participants excluded (e.g., Kruger, Hefer, & Matthew, 2013; Kang, Kweon, & Choi, 2022). Tracking loss is defined as the ‘the proportion of time that the eye tracker recorded point of gaze coordinates over the entire task’ (Amso, Haas, & Markant, 2014, p. 2).”

Amso, D., Haas, S., & Markant, J. (2014). An eye tracking investigation of developmental change in bottom-up attention orienting to faces in cluttered natural scenes. PLoS ONE, 9, e85701.

Kruger, J. L., Hefer, E., & Matthew, G. (2013). Measuring the impact of subtitles on cognitive load: Eye tracking and dynamic audiovisual texts. In Proceedings of the 2013 Conference on Eye Tracking South Africa (pp. 62–66). DOI: 10.1145/2509315.2509331.

5.1.1.

d > 0.001 ??? � d < 0.001

Response: 

Thank you for your correction. We changed as suggested in the revised manuscript. 

p. 13

“An independent-samples t-test was performed to analyze whether the time…”

This exact wording is too repetitive.

Response: 

Thank you for your due comment. We varied the wording in the revised manuscript. 

Results � The results

Response: 

Thank you for your detailed comment. We changed as suggested in the revised manuscript. 

p. 15

The 2nd and the 3rd paragraphs are very repetitive and redundant. The authors would want to focus on interpreting the results in this section.

Response: 

We revised the 2nd and the 3rd paragraphs as following: 

“The first research question proposed: What are the effects of a test announcement on learners’ attention? As mentioned earlier, the current study used eye-tracking to investigate participants’ attention. The eye-tracking results showed that there was a significant difference between TA and NTA groups only in processing bottom-margin target words. In other words, the TA group spent significantly more time processing bottom-margin target words than the NTA group for all three eye-tracking measures (i.e., gaze duration, second-pass reading time, and total reading time).”

Plus, there are studies that used eye-tracking technology to examine if glossing works.

Response: 

To the best of our knowledge, studies that used eye-tracking technology to examine if glossing works are limited to are limited to Kang et al. (2020), where reading English texts were examined, and Montero Perez et al. (2015, 2018), in which viewing video was examined. 

Kang, H., Kweon, S. O., & Choi, S. (2020). Using eye-tracking to examine the role of first and second language glosses. Language Teaching Research, 26, 1252-1273.

Montero Perez, M., Peters, E., & Desmet, P. (2015). Enhancing vocabulary learning through captioned video: An eye-tracking study. The Modern Language Journal, 99, 308-328.

Montero Perez, M., Peters, E., & Desmet, P. (2018). Vocabulary learning through viewing video: The effect of two enhancement techniques. Computer Assisted Language Learning, 31, 1-26.

Were the target words necessary for comprehension? The authors would want to consider the task-essentialness of the target words. That is, if the target words didn’t matter (especially adjectives or adverbs), they could have felt only little need (Laufer and Hulstijn’s Involvement Load Hypothesis) to know the meaning of the target words.

Response: 

We assume that the need, namely at least one aspect of motivation in Laufer and Hulstijn’s (2001) construct of task-induced involvement load, was not very different among different part of speech. Thus, we thought that adjectives and adverbs could play an role as important variables to comprehend the content of the text, even though it is possible that they may be relatively less important than nouns or verbs. That is why we controlled part of speech of the target words equally. 

p. 16

The difference between these results and ours might have occurred because recall and recognition (i.e., of forms) involve different cognitive efforts.

Please elaborate more on this.

Response: 

We added the following passage in the revised manuscript. 

“Unlike the findings of the current study, the lack of significant difference between the TA and NTA groups might have occurred because Ko (1995) did not examine form acquisition of vocabulary. Peters (2007) used a form recognition test as opposed to a form recall test, which was used in the present study. Furthermore, the difference between these previous and current results might have occurred because recall and recognition (i.e., of forms) involved different cognitive efforts. In other words, recognition test required choosing an answer among many options and allowed a chance effect, thus, involved relatively fewer cognitive efforts than recall test, where no clue was provided. It is possible that the difference between TA and NTA groups will be greater for tests in which relatively more cognitive efforts (i.e., recall) were required than for tests in which relatively fewer cognitive efforts (i.e., recognition) were needed.”

Our results reveal that vocabulary test announcement can affect which type of vocabulary

knowledge can be acquired. 

I'm not sure if I can understand this part. The authors may want to revise it.

Response: 

We revised as following: 

“The current results revealed that the effect of vocabulary test announcements can vary according to the kind of vocabulary test: form recall vs. meaning recall.” 

That is, test announcement can help the students to encode forms of new words, but that additional cognitive resources will be required to map forms to meanings of encoded words, or to consolidate form-meaning association of the encoded words.

Glossing in and of itself, promotes learning of word meanings. Previous studies overall showed that glossing promotes learning of word meanings, rather than forms. To be more specific, when glossing is not provided, readers tend to inspect the text more intensively to infer the meaning of the unknown word, especially if the word is important in comprehension. When glossing is provided, they do not feel the need to search for the meaning, as the gloss is right there, and hence have no or even negative impact on learning word forms. Perhaps additional demands induced by the TA could have compensated this relatively weak/inhibitory impact of glossing on learning of word forms. The authors should review more studies on the impact of glossing on learning of word forms, and not to mention, Laufer and Hulstijn's Involvement Load Theory. w

Response: 

We understand your concern about the role (effects) of ‘the need to search for the meaning’ in lexical acquisition. However, please understand that glosses are given (therefore, controlled) in all conditions, suggesting that the present study does NOT address glossing effects. Rather, the present study focuses on the relative efficacy of TA and NTA on (a) learning of visual word forms and (b) fostering form-meaning connections. For instance, we tested whether ‘glosses in the TA condition’ may be superior to ‘glosses in the NTA counterpart’ to acquiring word forms, not to promoting form-meaning links.

We presented fruitful findings on and implications of test announcement prior to reading a

glossed text in vocabulary acquisition.

The authors would want to revise this sentence, especially the word choice "Fruitful."

Response: 

We changed as following in the revised manuscript. 

“The findings of this study underscored the implications of test announcements prior to reading a glossed text in vocabulary acquisition”

Information about the type of vocabulary test might have changed the type of vocabulary that the participants learned.

I am not very suer if the word "type" is recommendable. The authors would want to reconsider this term throughout the manuscript to enhance clarity.

Response: 

We changed the word “type” into proper forms throughout the paper, such as “kind” of vocabulary test, or “aspect” of vocabulary knowledge. 

The authors would want to use the past tense when reporting what they DID or FOUND in this study throughout the methodology, results, and discussion sections.

Response: 

We did as the reviewer suggested. Thank you very much!!

Reviewer 3

Comments on “Effects of vocabulary test announcement prior to reading a glossed text on reading behaviors and vocabulary acquisition: An eye movement study”

The paper is in general well written. The research was carefully designed and the use of eye-tracking method is innovative. There are a number of issues, however, that need to be considered by the researchers.

1. Concerning form / meaning distinction

It is good that the researchers, based on previous studies, went a step further to measure the effect of test announcement on form recall and meaning recall. The researchers, however, did no review literature about the form / meaning distinction in L2 vocabulary learning. Since form / meaning are key constructs in the research, there need to be more discussions about them in the literature review. 

Response: 

We added the explanation of form-meaning distinction in L2 vocabulary learning as below in the revised manuscript. 

“The in-text target words are presented in the text and the bottom-margin target words are occurred at the bottom of the text in L2. Both correspond to the ‘form’ of a word which can also be called morphological information such as spelling or pronunciation (Jiang, 2000). When learners look at the in-text or bottom-margin target words, they may acquire the orthographic lexical form of the words (i.e., fusk, tarb). On the other hand, glosses enable learners to obtain semantic information, that is the ‘meaning’ of words (i.e., 소음, 가방). It may encourage learners to activate the form-meaning links (i.e., fusk-소음, tarb-가방). Thus, the reading time that the participants spent gazing at the ‘in-text target words’ and ‘bottom-margin target words’ can be related to the acquisition of ‘form’ of words, and the time spent reading the glosses can be related to the ‘form-meaning connection’ of words.”

2. In the research, there are three ROIs. What are the reasons of setting three ROIs for each target word? Please state the rationale.

Response: 

We added the following in the revised manuscript. 

“We set an ROI for each of three words: in-text target word, bottom-margin target word, and gloss, respectively.” 

Please understand that we did NOT set three ROIs for each target word. Rather, we set an ROI for each of three words (1) In-text target word, (2) bottom-margin target word, and (3) Gloss, respectively. 

Isn’t the bottom margin-target word a part of the gloss?

Response:

We defined a gloss as a definition or short explanation of an unfamiliar word in a text. Thus, we differentiated a bottom-margin target word (i.e., hule) from a L1 gloss (i.e., 가득찬). Also, when a participant looks at a gloss, she looks for the meaning of the target word, not its form. Thus, we set different ROIs for a bottom-margin target word and a gloss. Of course, a target word should be presented with its meaning (i.e., gloss), otherwise, we do not know of which word a gloss is. Mostly, glosses are presented this way in textbooks and many reading materials. 

3. Why are there F2 words and F4 words? The frequency of the target words is not a variable in this research.

Response: 

As we first started the current research, we wanted to see the frequency effects of the word as well. However, we determined not to see the frequency effects, because the conditions of the words of two frequency groups were not controlled in terms of different contexts and distributions.

4. What are the purposes of carrying out a meaning recall test and a meaning recognition test? Please justify this.

Response: 

We added the following in the revised manuscript. 

“We used two different form-meaning association tests to examine whether test announcement can affect the mapping of form to meaning differently according to different tests. More specifically, the meaning recall test measures relatively stronger form-meaning association by asking to recall a meaning of a given target word, rather than the meaning recognition test that allows to choose the best answer among multiple-choices.”

5. Was the form recall test more like a memory test instead of a vocabulary learning test?

Response: 

Yes, it was. Please understand that learning new words basically entails memory of novel word forms (Jiang, 2000). Previous research on vocabulary acquisition conducted the meaning recall test as we did. 

6. The scoring method for the meaning recall test can be improved since it seemed to be more flexible compared with the form recall test. The researchers are encouraged to try: 0 – incorrect; 1 – partially correct; 2 – correct.

Response: 

We understand your concern. However, please understand that most glossing studies utilized the same scoring methods as in our study. Basically, we considered form-meaning association was successfully done when a participant provided a synonym of the target word. Thus, we scored a partially correct answer (synonym) as correct answer (the same word). Please understand that this binominal method of scoring did not hinder examining participants’ acquisition of meaning of words. 

7. Another issue is about the reading comprehension questions. Are they related to the target words? Explanations are needed in the manuscript.

Response: 

We added the following in the revised manuscript. 

“In the reading comprehension test, some questions were related to the target words. For example, in order to answer a particular question asking which character did a specific behavior in which season, a participant needed to remember the form and meaning of a target word ‘marve’ meaning ‘봄’ (spring in Korean) presented as an option out of four choices. For a reading comprehension test item in which ‘marve’ is an answer, see question 5 in Appendix I.”

8. Are the target words important for the comprehension of the text? This is to be explained in the research design.

Response: 

We added the following in the revised manuscript.

“Both sets of target words consisted of four nouns, four verbs, four adjectives, all of which were content words. The fact that all target words were content words might have facilitated participants’ accurate and fast comprehension of the text. However, it seems to be impossible to objectively measure to what extent they are important.”

9. Test announcement is one step to initiate L2 learners’ noticing of the target word in reading. Whether the words are retained or not depends largely on the learner’s subsequent processing of the words, esp. depth of processing. As the researchers reviewed in the manuscript, a number of studies have been done on the effect of test announcement on L2 vocabulary learning, it is suggested that in future research the researchers focus more on processing depth and the mechanisms leading to differential learning outcomes of word form / meaning.

Response: 

We added the following in the revised manuscript. Thank you for the suggestion. 

“As a reviewer suggested, further research should focus on L2 learners’ depth of processing and the mechanisms leading to differential learning outcomes of word form and meaning to see the effect of test announcement”

---

## [Editor Report · Decision Letter 1]

4 Jan 2023

Effects of announcing a vocabulary test before reading a glossed text on reading behaviors and vocabulary acquisition: An eye-tracking study

PONE-D-22-15213R1

Dear Dr. Kweon,

We’re pleased to inform you that your manuscript has been judged scientifically suitable for publication and will be formally accepted for publication once it meets all outstanding technical requirements.

Kind regards,

Kathy Conklin, PhD

Academic Editor

PLOS ONE
---

## [Editor Report · Acceptance letter]

9 Jan 2023

PONE-D-22-15213R1 

Effects of announcing a vocabulary test before reading a glossed text on reading behaviors and vocabulary acquisition: An eye-tracking study 

Dear Dr. Kweon:

I'm pleased to inform you that your manuscript has been deemed suitable for publication in PLOS ONE. Congratulations! Your manuscript is now with our production department. 

Kind regards, 

on behalf of

Professor Kathy Conklin 

Academic Editor

PLOS ONE